# The Architectural Terracotta Marks of *Bracara Augusta* (Braga, Portugal): A First Typology Classification

Jorge Ribeiro [1,*] , Filipe Antunes [2,†] and Ana Fragata [3]

1 Lab2PT/IN2PAST, Institute of Social Sciences, University of Minho, Campus de Gualtar, 4710-057 Braga, Portugal
2 Museum of Archaeology D. Diogo de Sousa, Rua dos Bombeiros Voluntários, 4700-025 Braga, Portugal
3 GeoBioTec, Geosciences Department, University of Aveiro, Campus de Santiago, 3810-193 Aveiro, Portugal; afragata@ua.pt
* Correspondence: jribeiro@uaum.uminho.pt
† This author had passed away.

**Abstract:** Architectural Terracotta (ATC) is one of the most common materials in excavations from the Roman period. These ceramic building materials are an essential component of construction. Some of these pieces show potter´s marks, of different categories, that allow access to the production world of these materials. This investigation is a first typological classification of the 1216 marks from ATC materials, collected from 41 archaeological sites in *Bracara Augusta* (Braga, Portugal). Most of the marks were collected from the *domus* of Carvalheiras, one of the most emblematic archaeological sites of the city, currently under a musealization process. With this work it was possible to correlate the studied marks with specific terracotta types (shapes), context distribution and associated chronologies. The results suggested an organized and dynamic production, and an open-market, supported by numerous *officinae*, certainly of different sizes. Some of them were located near the housing area and reveal the presence of a large number of workers, including women and children. Further approaches on mineralogical, chemical and technological characterization of ATC, linked with stratigraphy, are under development.

**Keywords:** *Hispania*; Roman Western provinces; *Bracara Augusta*; tilery; tile; brick; ATC; marks

## 1. Introduction

French archaeologists that study the Roman and Medieval periods use the term "ATC: Architectural Terracotta (ATC)", to designate roof tiles (*tegulae* and *imbrices*), bricks, pipes, circular, semicircular and quarter-circle bricks, and other elements associated with the building of thermal spaces [1]. These ceramic building materials, such as tiles and bricks (Figure 1), had an important role in the construction of Roman cities, being used in multiple contexts, particularly in roofs and thermal buildings. However, unlike other types of ceramics, they have been unexplored in academic works.

*Bracara Augusta* (Braga, Portugal) was an economically and politically important city in the Iberian Peninsula, founded by the emperor Augustus in the late first century BC. (Figure 2). It was the capital of the *conventus bracaraugustanus* and, in the late third century AD, became the capital of the new province of *Gallaecia*, under emperor Diocletian. During the fifth and the sixth centuries the city was the capital of the Suebi Kingdom, which was absorbed and annexed by the Visigoth Kingdom in AD 585. From then on, and until the end of the seventh century, the city lost some political importance, but not religious, with the leading role being driven by the Church [2–5].

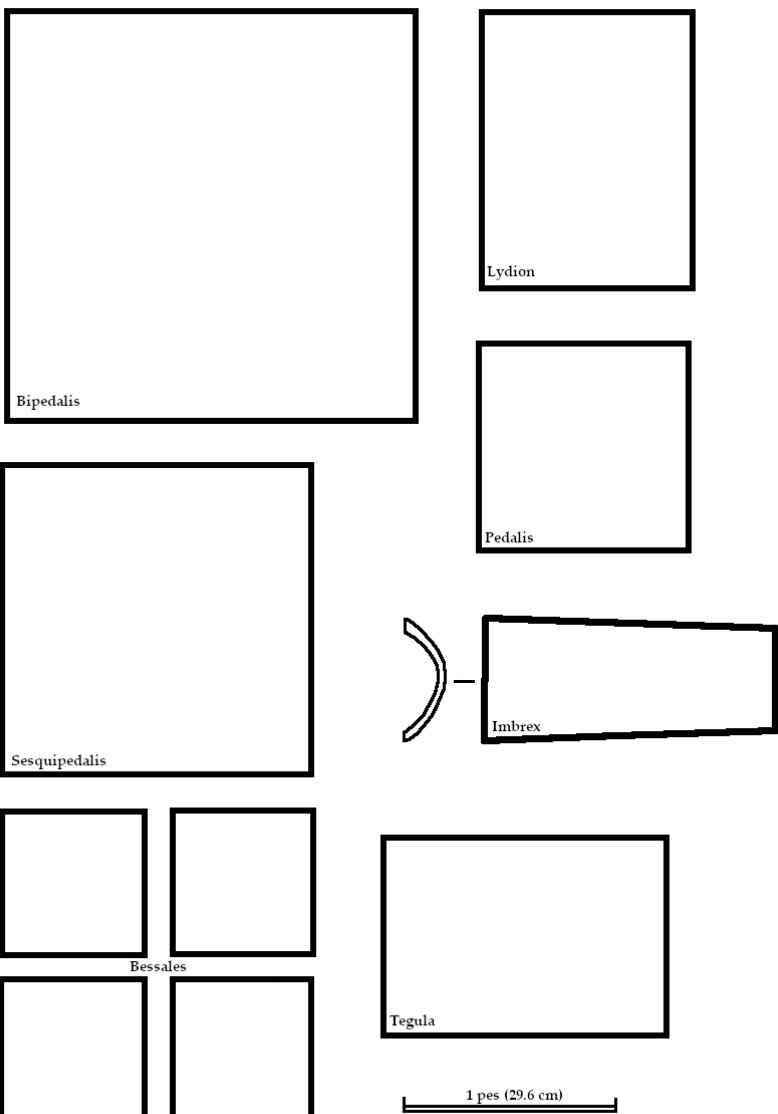

**Figure 1.** Brick and tile typology by Brodribb.

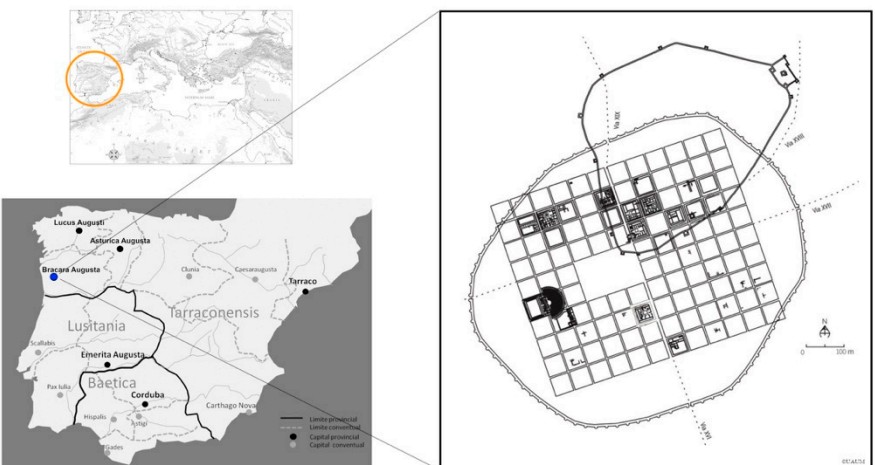

**Figure 2.** Location of *Bracara Augusta* in *Hispania* and plan of the city/low-imperial (around the city grid) and medieval walls, with the castle to the northeast (Modified from Magalhães [6]) (©UAUM).

The production of ceramic building materials in *Bracara Augusta* seemingly had an important social and economic impact. Some of these pieces showed marks that, if studied in more detail, could provide a better understanding of the actors involved in ATC production and some aspects related to their organization. In Braga, three types of marks were globally identified: stamps, finger-made marks and *graffiti* (alphabetical and numeral marks). This study presents a first typological classification of these marks, taking into account archaeological and historical evidences. Thus, 1216 marks were catalogued, from 41 archaeological sites of the city and from buildings that had different functionalities, from the foundation of the city in the first century BC, until the seventh century AD.

In this article, we considered other similar studies carried out in other contexts of the Roman world, namely in France [1,7], where various investigations focusing on the different type of marks under analysis provide elements of comparison for the Western Roman World. We approached several questions related to the various marks, especially regarding their function, position and orientation, in order to identify different *officinae* and their role in the organization of production and the techniques used in the manufacture of materials. A first report on the study of these marks was carried out by Filipe Antunes, a former technician at the Museum of Archaeology D. Diogo de Sousa (MDDS), in Braga. All marks were analysed and classified, from historical, critical and technological perspectives.

## 2. The *Laterarii*'s Marks, Their Interest and Study

ATC materials occur in large quantities in the archaeological record and our approach results from the valuable information that can be extracted from this material, related to their great typological variability and marks.

In previous studies, Goulpeau and Le Ny [7] investigated the fingerprint marks on Gallo-Roman construction baked clay materials and found that almost all types of elements from archaeological contexts have marks intentionally placed by the *laterarii*, perhaps with the exception of *tubuli laterici*. There are four major families of marks [8]. Finger-made marks, called signatures in English archaeological studies [7], are usually the most used, corresponding to simple or more complex lines, drawn on the surface of the piece. *Graffiti*, show the greatest variability, from simple letters, combinations of letters (representing initials of names), complete sentences and also numeral symbols. Stamps (*signacula*), a very rare type of mark, are made with a matrix in metal, ceramic or wood, with the name of the potter or owner [9]; and incisions, as lines representing Roman numbers, are usually made on one of the sides of the pieces.

These marks can have different meanings: the signature and *graffiti* can be related to the *officinae*, the stamps with the distribution of the materials, and incisions can be related to counting pieces. As stated by Charlier, this was the only way to explain that some pieces

had three different types of marks, as was the case of a *tegula* found in Switzerland with a finger-made mark, a stamp and an incision [8].

The main objective of the study of marks present in ATC was to investigate the *laterarii* society in terms of manufacturing processes, work organization, such as the number of workers or teams of workers involved, the number of *officinae*, or even the circulation of production. Additionally, important information on the potter's manufacturing techniques, age or gender, could be based on the size of the fingers, and accidental marks could be associated with women and child labor.

The study of signatures has benefited from numerous contributions, particularly in the work of Swiss and French archaeologists in the first half of the 20th century [10–12]. Only 20 years later, in 1973, new developments emerged, with Neumann's publication on Vindobona building materials [13], and above all Brodribb's intense work on Beauport Park (GB) materials [14], updated in 1987 [15]. Two years later, in 1989, an article dedicated to finger-made marks and incisions in Roman Gaul was published [7]. For the issue of *graffiti*, the contribution of Charlier is significant, who in 1999, investigated the marks in the context of socio-legal conditions of the *officinae* work [8] (work in which the author addressed the various types of marks), as well as on the practice of writing in the tiled areas of Roman Gaul [16]. Stamps are the most studied type of marks. Among the studies concerning the reality of the different Roman provinces, we can mention the publication by De Poorter and Claeys [17], on the marks from Belgium, and the work of Rico dedicated to *Betica* production sites, their surroundings, and stamps [18]. The Ferdière investigation [1] was dedicated to the potteries in *Lugdunensis* and *Aquitania*, where the author elaborated a reflection on this type of mark. In the context of the South Mediterranean Provinces, we should give a special mention to the work of Mills, on the trade of ceramic building materials from Carthage and Beirut [19].

Concerning *Hispania* and, in particular, the territory that became Portugal, this topic was relatively unexplored [20–25]. In the specific case of Braga, it is important to stress the work of Morais [9] on the marks and *graffiti* identified in the city, which, although of a more generic nature, provided data for several publications under the scope of ceramic building materials [26,27]. From the same territory, the *Gallaecia*, it is also important to point out the work on the ceramic construction material of the Roman villa of Toralla [28], the marks on the building material of the Roman camp of Ciadella [29], and additional synthesis work on the building ceramics of this province [30].

## 3. Marks Meaning, Criteria and Marking Frequency

The main idea underlying this study was to approach the evidence of a relationship between marks and economic and social aspects. The researchers that have been investigating this theme are quite unanimous in considering that, although it is a field to be explored, the function(s) of these marks will certainly be related to the internal organization of the *officinae* [1,7], and thus, are a way to better understand the potteries' workers, mainly potters, molders and owners of the workshops. It should be noted that the marks reveal little or no information about some workers, namely those linked to minor tasks. One of the most consensual interpretations on the meanings of the marks—which could be composed of simple signs, a single letter, or several—may be associated to the day laborer, with implications on the daily production of each worker, and their payment or not. Therefore, the smaller *officinae* did not need many marks, as each individual mark would be associated with a worker. In this regard, a very interesting aspect could be found in the Edict of Diocletian, which mentions the payment according to a quantity of materials produced and to the task they did.

Several investigations, some more extensive than others, lead some authors such as Goulpeau and Le Ny to adopt a more cautious approach, stating that "there will not be a fixed practice, corresponding to the use of these marks for each workshop according to its own functioning internal logic" [7].

Regardless of the issues mentioned above, the analysis of these specific marks in some ceramic building materials suggest that the two main marking criteria seemed to be the speed, by using mainly simple lines, and easiness, through the use of curvilinear and rectilinear motifs [7]. The morphology of some marks does not seem to meet these standards, by using a combination of various types which may disclose the originality of some workers.

The finger-made marks were the first marks to be placed on the pieces, still on the worktable, right after the molding process and while the clay was still in a fresh state [8]. They were usually done with the fingers, or with the branch of a bush. In the different studied contexts, there was the use of single fingers from both right and left hands. The marks would, by rule, be created on the face of the piece facing the molder, in its working position. On the other hand, stamps were applied at a later stage in the process, when the clay was still drying and showing a consistency neither too soft nor too hard.

Previous studies disclosed that the frequency of the marks varied according to the type of material and the size of the production [7]. Thus, the production of *tegulae*, in part because of its multiple uses, has always been more important than the production of *imbrices*. Another paradigmatic example were the *pilae* from *hipocaustum*, which normally consisted of a set of 11 *lateres*. On the other hand, circular bricks, compared to the previous ones, were produced in smaller quantities. Additionally, it seemed that not all elements were marked.

## 4. Sample

This sampling resulted from the materials collected from the archaeological excavations that have been developed in Braga since the 1970s, in the context of the rescue project of *Bracara Augusta*, by the Archeology Unit of the University of Minho (UAUM) [31]. It should be noted that the marks considered in this study come from the places of use and not from manufacturing contexts, even though four kilns have been documented in or around the city. The only example of a fully excavated kiln, the activity of which was exclusively for firing *tegulae*, was identified in 2009, during the construction works of the new Hospital of Braga (excavation under the responsibility of archaeologist David Mendes). Moreover, other *officinae* were documented, including those located in the archaeological area of Casa do Poço [32,33], and the kilns discovered in Rua dos Falcões (numbers 8–10—Irmandade de Sta. Cruz), an intervention by the archaeologist Armandino Cunha from the Archeology Office of Braga City Council (unpublished). In addition, remains of other roman kilns were also found in the Avenida da Imaculada Conceição (Oficinas da Livraria Cruz), an intervention directed by UAUM (unpublished), and in Rua Santos da Cunha, this one identified in the context of the opening of the street in 1955 [34–36] (Figure 3).

The dataset gathered from these sites was very significant and diversified, amounting to 1216 marks from more than 40 contexts/archaeological zones (Table 1), with all types of bricks showing marks, except *tubuli* (Table 2). These marks belong essentially to the categories of finger-made marks (line, curve and derivatives), *graffiti* (almost all of them traced with the finger; one letter or more and numerals), and stamps. However, this sampling showed some limitations, since it dealt with different contexts of use, corresponding to different *officinae* and chronologies, between the first century and the seventh century AD. Comparing this with other studies, in a work carried out in the Armorica region, in High Britain, there were 1049 finger-made marks, ready to be analysed [7], and in England, Brodribb counted 186 marked elements, within hypocaust bricks and rectangular bricks, based on a total of 1167 valid elements [14].

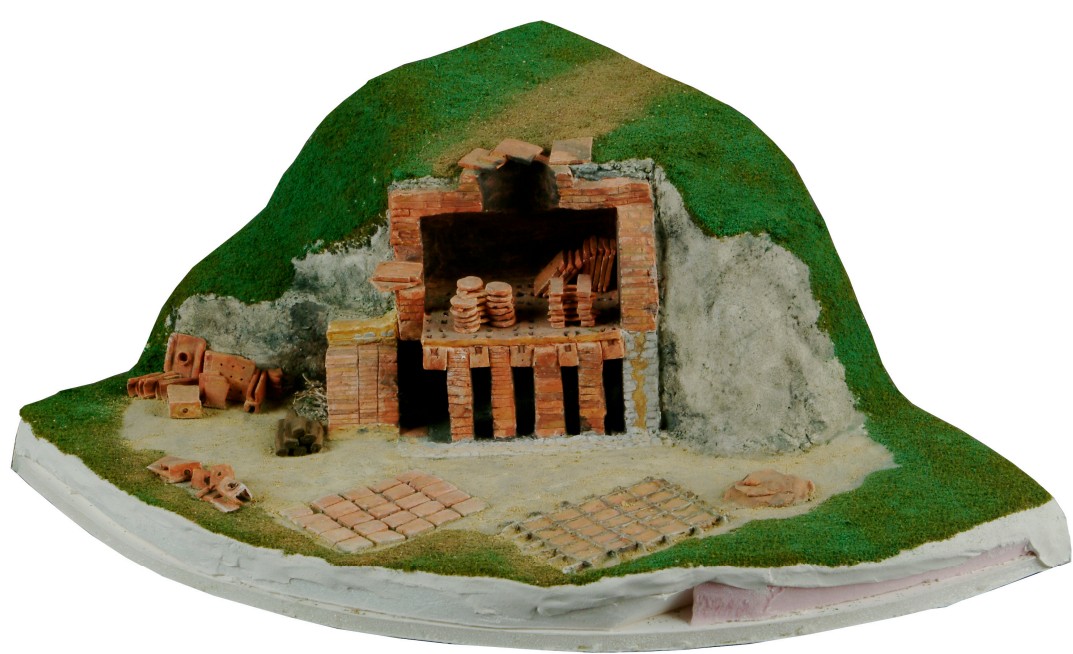

**Figure 3.** Model of tile kiln, by Filipe Antunes (©MDDS).

**Table 1.** *Bracara Augusta*: list of marks collected.

| Archaeological Site | Number of Identified Marks | % |
|---|---|---|
| Falperra | 23 | 1.89 |
| Dume | 15 | 1.23 |
| Braga (without context) | 98 | 8 |
| Paço | 1 | 0.08 |
| Muralha | 2 | 0.16 |
| Praia das Sapatas | 17 | 1.4 |
| Gualdim Pais (1989) | 2 | 0.08 |
| Gualdim Pais (2000) | 1 | 0.16 |
| "25 Abril" | 9 | 0.74 |
| **Carvalheiras** | **209** | **17.19** |
| Granjinhos | 45 | 3.7 |
| Rua A. Henriques (1993) | 6 | 0.49 |
| Cavalariças | 43 | 3.54 |
| Fujacal | 17 | 1.4 |
| Avenida Central necropolis | 16 | 1.32 |
| Rodovia necropolis | 2 | 0.16 |
| Carlos Amarante necropolis | 11 | 0.9 |
| Cangosta da Palha necropolis | 31 | 2.55 |
| Colina da Cividade | 36 | 2.96 |

**Table 1.** *Cont.*

| Archaeological Site | Number of Identified Marks | % |
| --- | --- | --- |
| **Termas** | **340** | **27.96** |
| Café Avenida | 17 | 1.4 |
| Rua S. Geraldo, 34 | 25 | 2.06 |
| Largo S. João do Souto | 4 | 0.33 |
| Hospital | 3 | 0.25 |
| Ex Albergue Distrital | 20 | 1.64 |
| Casa da Bica | 26 | 2.14 |
| Tanque de Água | 2 | 0.16 |
| Rua do Anjo, 55 | 22 | 1.81 |
| Misericórdia (Mis 96) | 10 | 0.82 |
| Rua de S. Geraldo (1994) | 14 | 1.15 |
| Cardoso da Saudade (1993) | 18 | 1.48 |
| Frigideiras | 9 | 0.74 |
| Rua Frei Caetano Brandão, 183/185 e rua Santo António das Travessas 20/26 (1998/2001) | 14 | 1.15 |
| Seminário de Santiago (1996) | 38 | 3.13 |
| Jardim da Misericórdia (1998) | 11 | 0.9 |
| Misericórdia A (1999) | 11 | 0.9 |
| Jardim da Misericórdia (1996) | 25 | 2.06 |
| Fujacal (1997/1998) | 15 | 1.23 |
| Sé | 4 | 0.33 |
| Rua de S. Sebastião | 2 | 0.16 |
| Fonte do Ídolo | 1 | 0.08 |
| Rua de S. Geraldo, 27/31 | 1 | 0.08 |
| **Total** | **1216** | - |

**Table 2.** *Bracara Augusta*: list of marks collected, according to ATC typology.

| Typology | Number of Marks | % |
|---|---|---|
| *Tegula* | **290** | **23.85** |
| *Imbrex* | 8 | 0.66 |
| Vault brick | 46 | 3.78 |
| *Latere bessalis* | 116 | 9.54 |
| *Latere pedalis* | 20 | 1.64 |
| *Latere sesquipedalis* | 4 | 0.33 |
| *Latere bipedalis* | 17 | 1.4 |
| *Latere longum semi bessalis* | 2 | 0.16 |
| *Latere longum semi pedalis* | **190** | **15.63** |
| *Latere longum bessalis* | 4 | 0.33 |
| *Latere lydion* | **466** | **38.32** |
| Beveled brick | 1 | 0.08 |
| Circular brick | 4 | 0.33 |
| Other shapes/ indeterminates | 48 | 3.95 |
| **Total** | **1216** | - |

However, despite the large quantity of analysed samples, it should be considered that pieces from two groups, in particular, were present in very large numbers: the *thermae* of Alto da Cividade and the archaeological zone of the *domus* of Carvalheiras, resulting from the extensive archaeological interventions developed here.

## 5. Methods

The approach to the finger-made marks in this study followed the methodology proposed by Goulpeau and Le Ny [7]. Their investigation on these marks in the context of Gaul was fundamental to issues related to the marks' frequencies, functionality and typological organization. Among other aspects, the authors standardized the description of the finger-made marks, based on a specific terminology, exact form and size of each mark. They proposed a rigorous and innovative reference typology, organized into large families of marks and their subtypes, which we have adapted to our study context.

Regarding the other types of marks, namely incisions, stamps and *graffiti*, we focused on some studies developed by Brodribb [14,15], Le Ny [37], Charlier [8] and Ferdière [1], dealing, respectively, with the study of materials from *Britannia* and Gaul, which provides us with elements related to the characteristics of production, the marks´ function, and even classification hypotheses, in comparable contexts to the Western Roman Empire.

In the present investigation a protocol based on the one proposed by the ATC Network (Réseau TCA) [38], a French working group dedicated to the study of ceramic building materials, in which we take part, was developed, considering the materials collection, classification and storing specificities. This protocol is complementary to the above methodology, allowing us to establish the bases of the work upstream of the study of the marks themselves, and valuing the importance of the ceramic building materials.

One of the main transversal findings in common to all authors that have been investigating this theme, is the need to organize the main forms into large types, or in other words into typologies. Considering the exact shape of each mark and its multiple variations, we tried to define a set of large families into which they could be grouped (see detailed description of the tasks in Figure 4).

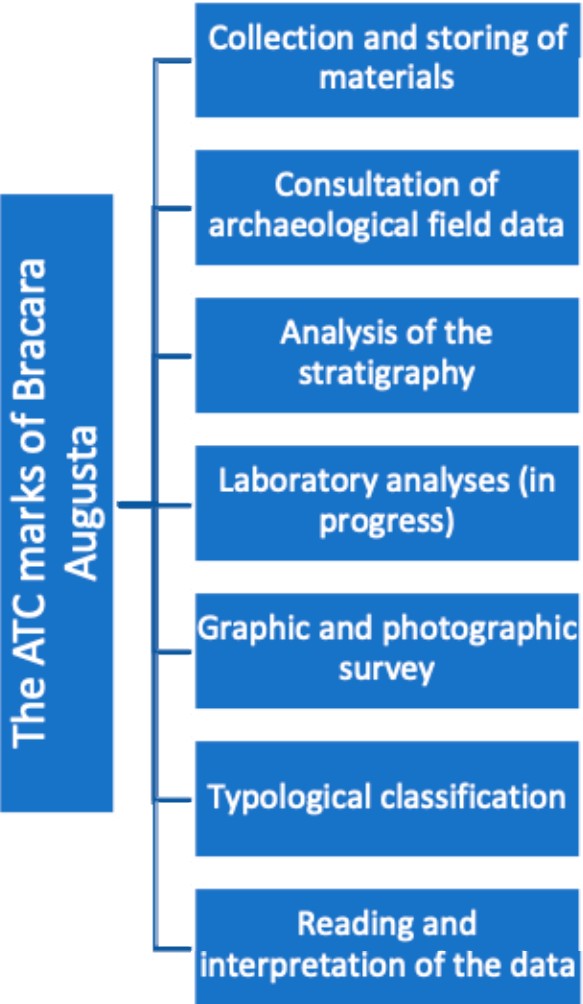

**Figure 4.** Main tasks of our study.

This approach presents some limitations, namely in the specific case of the finger-made marks and the *graffiti*, which are those that appear in larger numbers, due to the fragmented state of many marks, as well as to the great similarity of some motifs, which complicates the typological attribution to one or another family. Furthermore, the materials from the oldest excavations have not always been preserved, and their drawings do not have the necessary graphic quality.

This work should be seen as a first approach and its progression will have to go through updating of the typology and the study of the marks from the various referenced archaeological sites, namely those that are supported by a secure stratigraphy.

## 6. Results

We have established seven distinct groups: curvilinear—1, waves—2, loops—3, dashes—4, complexes—5 (finger-made marks), finger-line letters—6, finger-line numbers—7 (*graffiti*). Additionally, we created a group for (rare) stamps—8 and another one for accidental marks—9 (Figure 5). The *graffiti* group, which included finger-made letters and numerals, with 512 elements, was the one that assumed the greatest representation (42.1% of the total) followed, in decreasing order, by the curvilinear one (240 marks—19.74%), the dashes (217 marks—17.85%) and the loops (142 marks—11.68%) (Table 3).

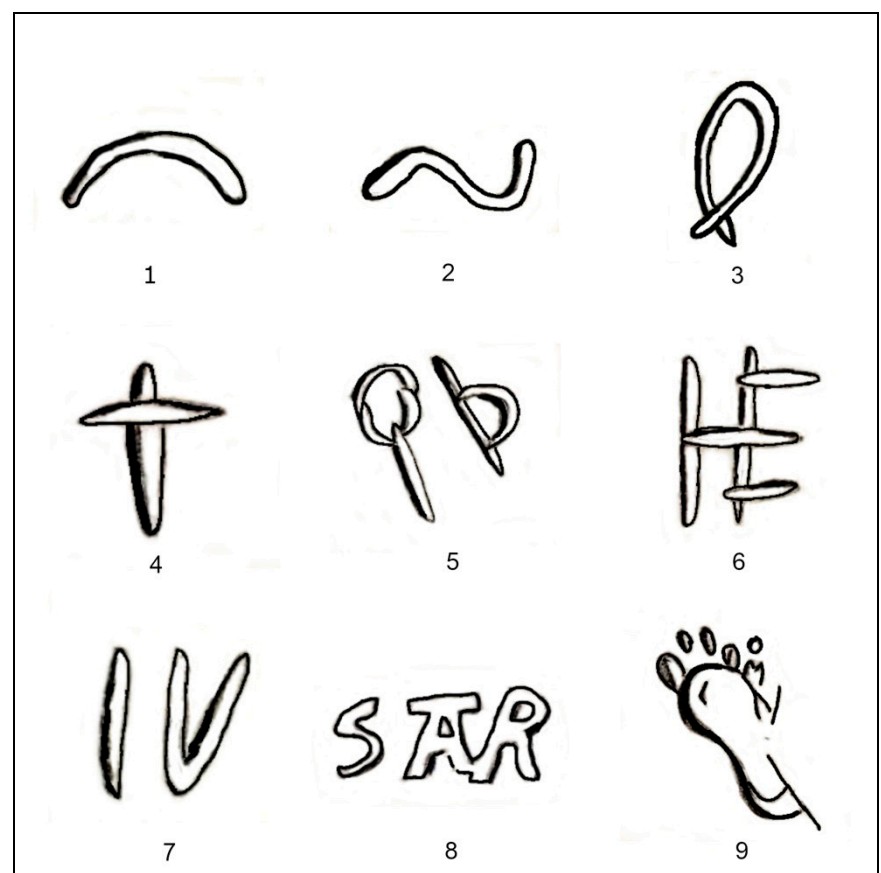

**Figure 5.** Families of marks: 1—curvilinear, 2—waves, 3—loops, 4—dashes, 5—complexes, 6—finger-line letters, 7—finger-line numbers, 8—stamps, 9—accidentals.

**Table 3.** *Bracara Augusta*: frequency of identified typology marks.

| Typology | Number of Marks | % | Group |
|---|---|---|---|
| Curvilinear | 240 | 19.74 | |
| Waves | 39 | 3.21 | |
| Loops | 142 | 11.68 | |
| Dashes | 217 | 17.85 | Finger made marks: 57.09 |
| Complexes | 47 | 3.87 | |
| Unclassified | 9 | 0.74 | |
| Finger-line letters | 453 | 37.25 | |
| Finger-line numbers | 59 | 4.85 | *Graffiti*: 42.1 |
| Stamps | 2 | 0.16 | 0.16 |
| Accidental | 8 | 0.66 | 0.66 |
| **Total** | **1216** | | - |

In the studied dataset, the marks that deserved the most attention were those associated with the names of the potters and information related to the production. Only two stamps were counted (belonging to the same potter): two *tegulae*, from excavations carried out in the Misericórdia archaeological site, and a follow-up carried out on the outskirts of Braga, marked with the acronym *SATVR* (Figure 6), which referred *Saturninus*, an owner or potter of one of the *officinae* that worked in the city [39]. This fact was in accordance with an investigation in Gaul [8] where only 15% of the workshops placed stamps on their

productions. It should be noted that a stamp of the same type, certainly from the same potter, was identified in a *tegula* collected in the hillfort of Lousada [25], located about 50 km from Braga. Despite the residual number of stamps collected in Braga, this situation reinforces the idea, defended in other studies, that this type of mark is, as a rule, associated with the distribution of materials to markets located outside of the city, suggesting open and dynamic markets. In addition, the studied marks revealed other names of potters, such as the genitives *PIRI* and *SABINI*, and the abbreviation *SIL* [9]. Those marks all came from the excavations of *thermae* of Alto da Cividade and were carried out with very fine lines, which could have been done with a stick. We believe that several of the *graffiti* highlighted, specifically the finger-line letters (*A*, *AE*, *AR*, *AT*, *CA*, *CE*, *CS*, *E*, *F*, *H*, *HE*, *N*, *NA*, *NN*, *P*, *R*, *RF*, *S*, *SE*, *SF*, *SS*, *ST*, *T*, *V*, *VA*, *VAF*, *VE*, *VP*, *VR*, *VV*) can also identify *officinae* or potters' names. Other local potters have been identified in lamps, namely *P. DOMITI* (*Publius Domitius*), *E[X?]MIC?*, *LVCRETI* (*Lucretius*), *OCTAVI* (*Octavius*) and *BASSI* [9].

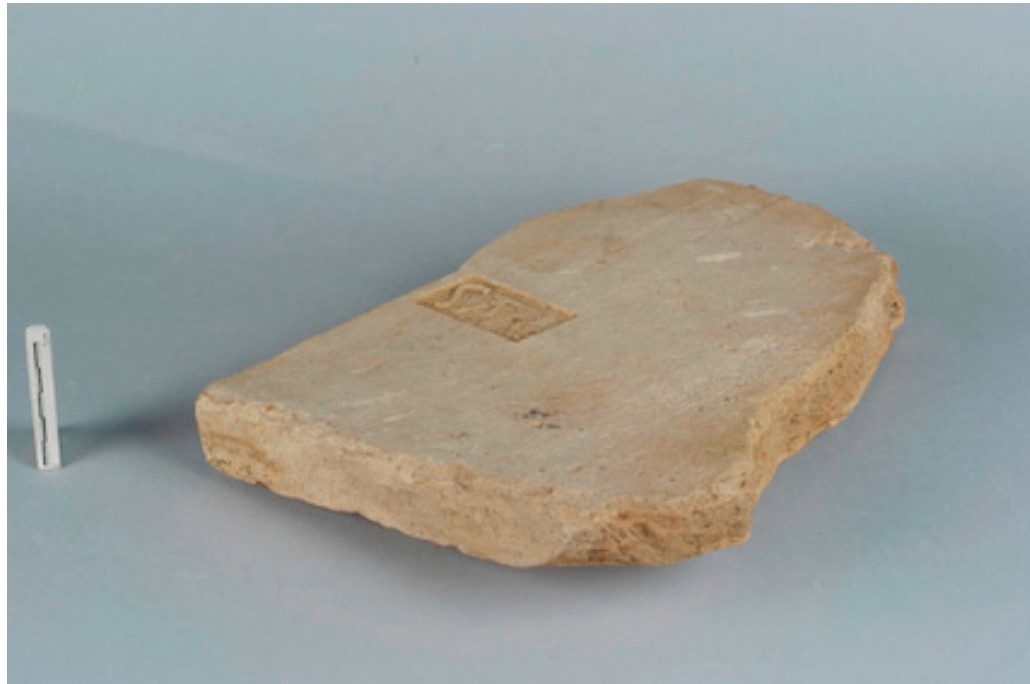

**Figure 6.** Stamp *SATVR* present on a *tegula*, from Misericórdia (©MDDS).

Two other marks, documented on *bipedal lateres*, were made with very fine lines, possibly using an object or a bush branch, and seemed to indicate the counting of the pieces in production. Those marks were collected in the Granjinhos area (Figure 7) and in the excavations of the *thermae* of Alto da Cividade. According to some authors [8], this kind of mark may be related to the type of contract that linked a worker to the person who requested the material, associated with the figure of *probatio*—receipt and validation of the ordered material.

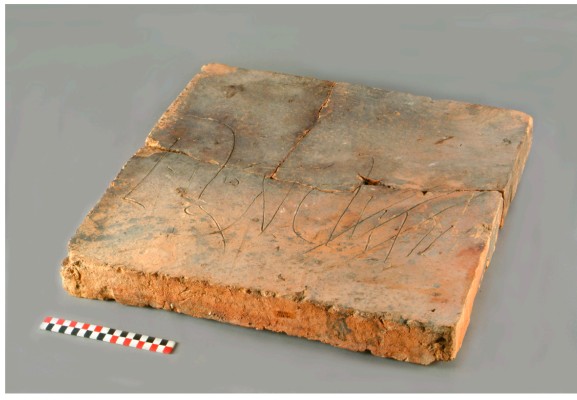 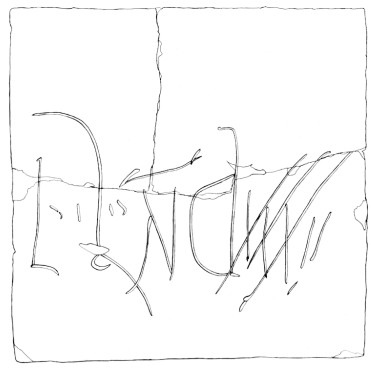

**Figure 7.** Mark of production counting, on a *bipedal* brick, from Granjinhos (© MDDS).

A large part of the marks displayed on the surface of the studied ATC (57.09%) are included in the group of finger-made marks, which exhibit a wide variety of shapes and combinations, and could have a particular role in the production of the materials. We think that these marks, dragged across the ATC material's surface, were made to allow the craftsmen´s product identification when picking them up after firing.

Some accidental marks were also highlighted. These include a female sandal mark and a female footprint from the archaeological zone of the *domus* of Ex Albergue Distrital, and Misericórdia necropolis, respectively (Figure 8). Both of these marks were identified in a previous investigation [40], suggesting the presence of children and women in the workshops, who could also have been potters. An animal footprint (6.2 × 6.94 cm), possibly belonging to a small dog, was also identified on a brick that formalizes the hypocaust area of the *thermae* of Alto da Cividade. Animal footprints are common on Roman ATC and suggest that the bricks/ tiles were left out to dry on the ground before firing. In this case, it probably means that the *officina* was located next to housing areas. These kinds of marks are interesting, since they tell us about the faunal diversity that accompanied human activities. In more rural spaces, for example, there are examples of pigs´ and goats' footprints [25].

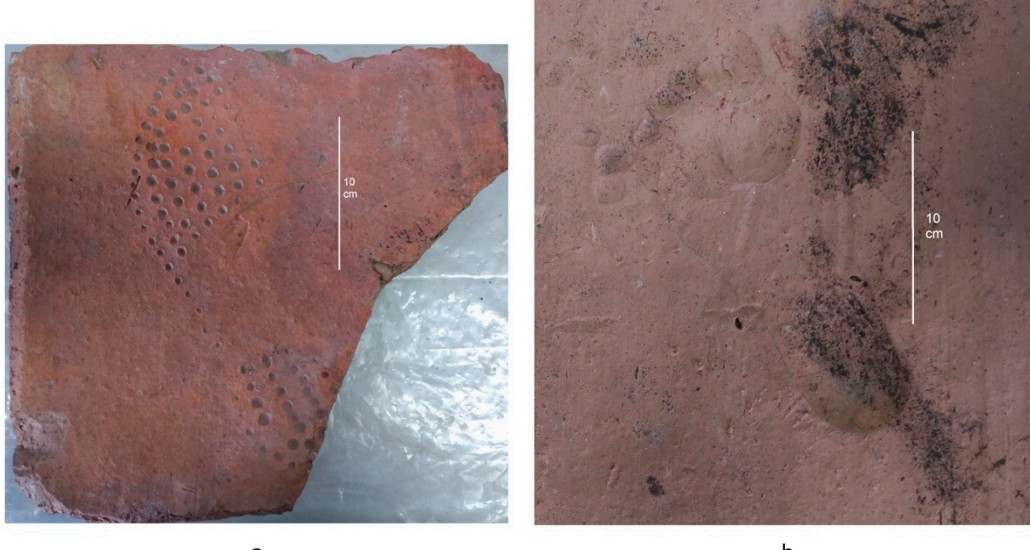

a                                            b

**Figure 8.** (**a**) Detail of shoeprint in a *pedale* or *lydion* brick from the *domus* of Ex Albergue Distrital. (**b**) Detail of footprint present in a *lydion* brick from the Misericórdia necropolis (adapted from Marado and Ribeiro [40]).

Another mark that caught our particular attention was the one with the initials "BA", represented in a quadrangular brick, coming from the archaeological area of Falperra (Figure 9), and collected in excavations carried out in 1968/69, in the vicinity of the building with a basilica plan that was identified there. This mark appeared to have been drawn with one finger, through six movements. This type of mark may reveal the participation of cities in productive activities. Similar marks occurred in Braga, namely in lamps, BAF–*Bracarae Augustae figlinis* (*Lucretius*) [9]. In the Portuguese territory, the municipal production of ATC was documented in *Seilium* with the mark *RPS* [*R(es) P(ublica) S(eiliensis)*], identified in two bricks, both performed by incision. The first one was on a brick of indeterminate shape and the second one on a *bessale* brick. For production in *Conimbriga*, the mark *RPC* [*R(es) P(ublica) C(onimbrigensis)*] was printed, in relief, on a brick, and in the settlement of Trêsminas, through the mark *AFL* [*A(quae) FL(aviae)*], inscribed on a *tegula* [24,41]. In other provinces, Fernandes and Ferreira [24] found that this type of reference was also quite rare, although the mark *CARTEIA* was printed on bricks collected in that city, and the mark *CIAE* [(*C(olonia) I(ulia) A(ugusta) E(merita)*)] was found in *tegulae* and lead pipes.

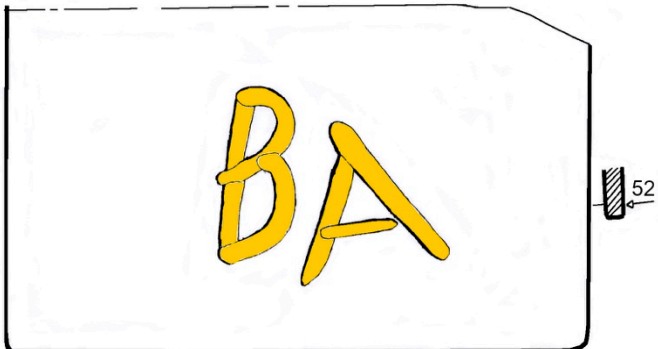

**Figure 9.** Graffito *BA* on a quadrangular brick from Falperra (adapted from ©MDDS).

*The Set of Marks from the Domus of Carvalheiras*

Considering the several archaeological sites where marks were retrieved, particular attention was paid to the set from the *domus* of Carvalheiras, an *atrium* and peristyle house, built at the end of the first century, and the reference for the study of high-imperial private architecture in *Bracara Augusta*. In the second century, the northern half of the house was completely transformed to build a *balneum*, with extensive use of ceramic building materials [4]. As such, the study focused particularly on the analysis of this dataset of conservable size and importance, in the context of the city´s domestic construction, but also due to the potential information embedded in the collected materials.

During the interventions carried out between 1982 and 2002, 244 bricks were recovered, with 209 marks, of which 98 were different from each other and 200 could be included within a typology (Table 4, Figure 10).

**Table 4.** *Domus* of Carvalheiras: list of ATC collected.

| Pieces and Marks | Number |
| --- | --- |
| Collected pieces | 244 |
| Marked elements | 209 |
| Marks with typology | 200 |
| Different marks | 98 |

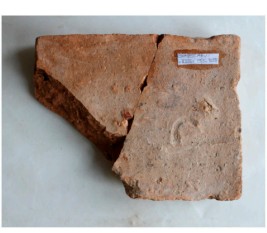
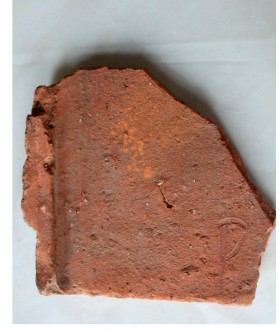
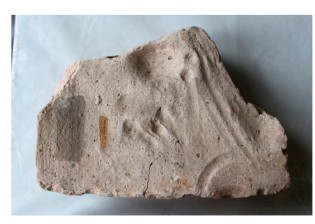
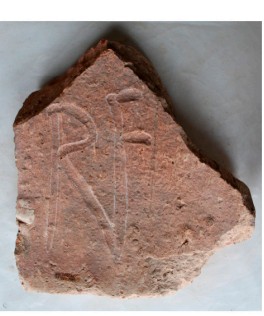
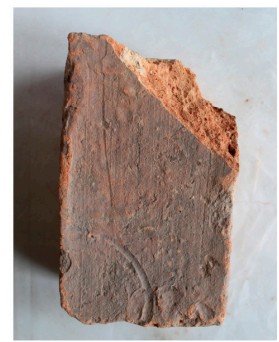
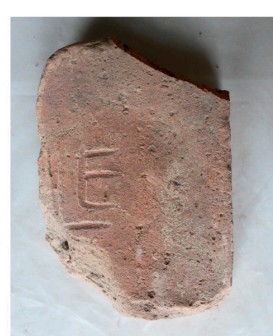

**Figure 10.** Set of marks in bricks from *domus* of Carvalheiras (pictures taken without scale) (©MDDS).

The proportion of marks in relation to the total collected set of bricks could be misleading, suggesting an extremely high frequency of marking, which was not the case, and could rather be explained by a logical selective collection of some shapes and pieces that displayed a mark. There were pieces of the most varied forms, included in the group of roofs, with 79 elements: 76 *tegulae*, distributed within seven distinct modules and three *imbrice*, and in the group of hypocausts of the thermal complex, with 40 pieces: 15 vault bricks, four beveled rectangular bricks, both possibly connected to the vaults, 20 *lateres bessales* (five of which wedge-shaped), two *sesquipedales*, equally with seven distinct modules, three *tubuli* and one *tegula mammata* (Table 5).

The majority of identified marks fell within the group of finger-made marks with 121 units (about 58% of the samples), followed by the *graffiti* group, with 78 units (about 38%). In the first group, the curvilinear ones stand out, with 64 units and 30.62% of the total, followed by the dashes, with 31 units making 14.83%. As for the *graffiti* group, they were mostly made of finger-line letters, with 68 units (32.54%), and a few finger-line numerals, with 10 units (4.78%) (Table 6).

Within these groups, the most frequent forms were the two downward-oriented circle arcs, usually located at the base of the piece, with 19 occurrences in different materials: *tegulae*, vaults bricks and *lydion* bricks; followed by the single wave, recorded 10 times, essentially in *tegulae*. In third and fourth place, appeared, respectively, two parallel lines and the simple circle arc, both with nine occurrences, the first in *tegulae* and *longum semi pedalis* bricks and the second in *tegulae* and *lydion* bricks (see Figure 11). In the set of *tegulae*,

47 distinct marks were documented, and only three in the vault bricks, a situation that may be explained by the specificity of this form, which would have a substantially lower number of potters dedicated to its manufacture.

In many types of marks, there were variations, which may have been associated with the "signature" of workers from the same *officinae* (Figure 12). In the context of *graffiti*, there were several marks documented that used one or two letter initials, often in a nexus, which refers to the names of potters, with similarities to other areas of the city (Figure 13).

**Table 5.** *Domus* of Carvalheiras: list of ATC collected according to the type, frequency and modules identified.

| ATC Type | Frequency | Modules |
|---|---|---|
| *Tegula* | 76 | 7 |
| *Imbrex* | 3 | - |
| Vault brick | 15 | 7 |
| *Latere bessale* | 5 | 4 |
| *Latere bessale cuneati* | 10 | 5 |
| *Latere sesquipedale* | 2 | 1 |
| *Latere longum sesquipedale* | 32 | 11 |
| *Latere longum bessale* | 3 | 2 |
| *Latere lydion* | 43 | 16 |
| *Latere* bevelled rectangular | 4 | 3 |
| *Tubuli laterici* | 3 | 2 |
| *Tegula mammata* | 1 | 1 |
| Pipes | 3 | 3 |
| Indeterminate | 44 | - |

| Mark typology | Frequency | ATC Typology |
|:---:|:---:|:---:|
| | 19 | *Tegula*<br>Vault brick<br>*Lydion* |
| | 10 | *Tegula* |
| | 9 | *Tegula*<br>*Latere longum semi pedalis* |
| | 9 | *Tegula*<br>*Lydion* |
| | 7 | *Latere longum semi pedalis* |
| | 6 | *Tegula*<br>*Latere longum semi pedalis* |

**Figure 11.** *Domus* of Carvalheiras: marks with higher frequency (adapted from F. Antunes sketch).

**Figure 12.** *Domus* of Carvalheiras: some of the identified variants of ATC marks (adapted from F. Antunes sketch).

| Mark | *Latere* | Other occurrences/ ATC type |
|---|---|---|
| | *Latere lydion* | Casa da Bica / LSP<br>Rua de S. Geraldo 94 / LSP<br>Termas / LSP<br>MIS A / LSP<br>MIS 96 / LSP |
| | *Latere longum semi pedale*<br>(LSP) | Falperra / ind.<br>Termas / *tegula* |
| | Indeterminate<br>(ind.) | N/A |
| | *Latere lydion* | N/A |
| | *Latere bessale cuneati* | Colina / ind.      Termas 98/99 / ind.<br>Praia das Sapatas / ind.      Fujacal / ind.<br>Termas *in situ* / ind.      T98/99 / *tegula* |

**Figure 13.** *Domus* of Carvalheiras: ATC marks in capital letters (adapted from F. Antunes sketch). (N/A: not applicable).

**Table 6.** *Domus* of Carvalheiras: frequency of identified typologies of marks.

| Mark Typology | Frequency | % | Subtotal % |
|---|---|---|---|
| Curvilinear | 64 | 30.62 | |
| Waves | 11 | 5.26 | |
| Loops | 7 | 3.35 | Finger-made marks: 57.89 |
| Dashes | 31 | 14.83 | |
| Complexes | 8 | 3.83 | |
| Finger-line letters | 68 | 32.54 | |
| Finger-line numbers | 10 | 4.78 | *Graffiti*: 37.32 |
| Indeterminates | 10 | 4.78 | |

## 7. Conclusions

*Bracara Augusta* was an important Roman city in *Hispania* and the analysis of the datasets of documented brick marks found there, in various archaeological sites, provided significant results and insights into the city´s brick/ tile workers and revealed part of the inner life of the *officinae*. In particular, the *domus* of Carvalheiras was an important case study due its size and diversity of available data. In this paper, we approached the ATC mark meanings, the marking criteria, the proportion of marked materials, and we proposed a typology essay. The analysis was based on a vast dataset of 1216 elements, within which we have individualized 847 distinct marks, which we classified into nine types. More than 90% of the marks collected belong to the group of finger-made marks (geometric shapes, finger-line letters and finger-line numbers). The types of ATC that are most marked are the *latere lydion* and the *tegula*, with respectively 466 and 290 examples. The documented evidence seemed to highlight an intense activity in the city, with the collaboration of many workers, at different periods, a fact proven by the stratigraphy documented in several of the archaeological sites mentioned, such as the *domus* of Carvalheiras. Similarly, the identification of variants in many of the documented types of marks could emphasize the presence of several potters in the same workshop. Certain elements revealed an organized production, with some workshops located, at least, at times, near residential areas, where men, women and also children seemed to work. The case of stamps was quite interesting, on the one hand, there were very few examples, with only two pieces found, on the other hand, they were usually associated with open and dynamic markets [7]. *Bracara Augusta*, partly due to its geostrategic position, was shown from very early times to be a privileged market, supplying the domestic consumption of the city and its area of influence, as proven by the study of its trade based on ceramic materials [9]. In the case of ATC materials, and reinforcing what was said above, we believe that a similar situation could have happened, although this aspect needs to be further explored. Some of the marks were associated with the workers that made them, identified by their names and they could be the owners of the workshops, the production foremen or even "simple" workers. The specific functionality of the marks and the marking criteria still need further investigation, but some of the cases recorded in Braga seemed to lift the veil on the internal life of the *officinae*, informing about the different responsibilities in production between the workers and the owners.

The main limitations of this study are related to the fragmentary nature of some marks, which makes their reading, interpretation and inclusion in one or another category difficult. In the same way, the pieces collected in the oldest excavations do not present a secure stratigraphic framework. It should also be noted that the majority of the marks identified in the city belong to the group of the finger-made marks and *graffiti,* precisely those which offer greater difficulty in terms of understanding their functionality.

Future work on the mineralogical and chemical characterization of ATC, under development and already applied by our team in the context of ancient mortars [42,43], could give more information on the raw materials used, namely their provenance, as well as the

manufacturing process, workshop organization and production distribution. These characterization analyses could be linked with stratigraphy, allowing a deeper contextualization, which might lead to the development of further chronotypologies. Additionally, it would be important to consider the development of an adequate analysis platform, defining, for example, the technical aspects (movements performed, fingers used, etc.) and metrological aspects and extending the comparative analysis to other areas of marked ATC. Another possible increment to this investigation could be the update of the initial database, with materials collected from excavations in recent years, that could result in the identification of groups of marks with greater frequencies and in the updating of known marks.

**Author Contributions:** Conceptualization, J.R. and A.F.; methodology, J.R. and A.F.; validation, J.R. and A.F.; formal Analysis, J.R. and A.F.; investigation, J.R., A.F. and F.A.; writing—original draft preparation, J.R. and A.F.; writing—review and editing, J.R. and A.F. All authors have read and agreed to the published version of the manuscript.

**Funding:** This research was supported by the Landscape, Heritage and Territory Laboratory (Lab2PT), Ref. UIDB/04509/2020, financed by national funds (PIDDAC) through the FCT/MCTES, and the Geo-BioSciences, GeoTechnologies and GeoEngineering Research Centre (GeoBioTec), Ref. UIDB/04035/2020, funded by FCT and FEDER funds through the Operational Program Competitiveness Factors COMPETE and by national funds (OE), through FCT in the scope of the framework contract foreseen in the numbers 4, 5 and 6 of the article 23, of the Decree-Law 57/2016, of 29 August, changed by Law 57/2017, of 19 July.

**Institutional Review Board Statement:** Not applicable.

**Informed Consent Statement:** Not applicable.

**Data Availability Statement:** Not applicable.

**Acknowledgments:** The authors wish to thank and remember Filipe Antunes, whose interest on brick marks at the Museum of Archaeology D. Diogo de Sousa, in Braga (MDDS) allowed the development of this approach. The collection and synthesis work (unpublished) developed by Filipe Antunes, former technician at the MDDS, who collected, inventoried and drew the different marks in materials collected from the excavations in Braga, until the beginning of the 2000s, outlined a first typology approach and constituted the basis of this study. The sampling was carried out at the MDDS. The authors are grateful to Isabel Silva (Director of MDDS), Maria José Sousa, Clara Lobo, Vitor Hugo Torres, Isabel Marques, Manuel Santos, Amélia Marques and all the staff of MDDS, for their help in the access to the materials of interest to this study. Manuela Martins, Luis Fontes, and other archaeologists at the Archaeology Unit of the University of Minho also provided access and context to the materials.

**Conflicts of Interest:** The authors declare no conflict of interest.

## Abbreviations

| ATC | architectural terracotta |
|---|---|
| GeoBioTec | GeoBioSciences, GeoTechnologies and GeoEngineering Research Centre |
| IND | indeterminate |
| IN2PAST | Associate Laboratory for Research and Innovation in Heritage, Arts, Sustainability and Territory |
| Lab2PT | Landscape, Heritage and Territory Laboratory |
| LSP | *longum semi pedalis* |
| MDDS | Museum of Archaeology D. Diogo de Sousa |
| N/A | not applicable |
| TCA | terres cuites architecturales |
| UAUM | Archaeology Unit of the University of Minho |

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
