# Peer review of "The Architectural Terracotta Marks of Bracara Augusta (Braga, Portugal): A First Typology Classification"

_heritage, doi:10.3390/heritage4040227_

Round 1

Reviewer 1 Report

- Lines 31-34: In the text, the historical contextualization of the city ends in the s. III A.D.. However, the chronological context of the samples reaches the late antiquity (5th c. - line 46). It is recommended to extend the historical contextualization to at least the 7th c. (capital of the Kingdom of Suebi, Metropolitan church, etc.).

  • This chronological problem is also detected if we observe the sampling sites. Two sites stand out over the others: Termas and Carvalheiras. These two sites have ceramics with chronologies that reach the beginnings of 6th c. And in two other archaeological sites like Falperra and Dume, the pottery contexts reach centuries later and even medieval. In fact, an important example of BA Graffito comes from Falperra = (Fig. 7). Therefore, it is proposed to revise the chronology of the sample in the text
  • Line 62: 2. The Laterarii´s marks, their interest and study
    • In the discussion on international studies I think P. Mills' work on Ceramic Building materials from Carthague and Beirut published by Archaeopress in its RLAMP collection (2013) should be mentioned. 

      https://archaeopress.com/Public/displayProductDetail.asp?id={C5CE327D-710F-44D9-AC3A-7C04BC75D091}

    • The authors refer to studies in Portugal and specifically in Braga (lines 110-117). It seems necessary to point out similar studies of the same territory, the Gallaecia. In the same conventus iuridicus stands out the work on the ceramic construction material of the Roman villa of Toralla (Vigo) 

      https://dialnet.unirioja.es/servlet/articulo?codigo=4158776

    • and some synthesis work on the building ceramics of the Gallaecia 

      https://dialnet.unirioja.es/servlet/articulo?codigo=7580812

    • or even works on the marks on building material of the Roman camp of Ciadella with references like: https://www.exofficinahispana.org/wp-content/uploads/2021/08/16_ciadella_rodriguez.pdf
  •  
  • Graphic scale should be incorporated into figure 4 and the brick drawing in figure 5

Author Response

Dear reviewer, please find attached the response to your comments.

Yours sincerly

Jorge Ribeiro

Reviewer 2 Report

1.      General Comments
In general, the manuscript is well written and organized. I am definitely in favour of publishing this manuscript. Nevertheless, the manuscript needs some further improved before to be accepted for publication. In general, there are still some occasional grammar errors through the manuscript especially the article ''the'', ''a'' and ''an'' is missing in many places, please make a spellchecking in addition to these minor issues. The reviewer has listed some specific comments that might be helpful of the authors to enhance the quality of the manuscript further. Please consider the particular comments listed below!

2.      Specific Comments

*       List of acronyms is needed.
2.1. Abstract
*       The abstract is well written. The structure is fine. Nevertheless, the concluding remarks should be added.

2.2.    Introduction
*       This section is well written; the objectives are explicitly stated.
*       The authors need to enrich further the background; in this regard, it is important to mention some international case studies:

Hemeda. S. 3D finite element coupled analysis model for geotechnical and complex structural problems of historic masonry structures: conservation of Abu Serga church, Cairo, Egypt. Heritage Science, 2019,7(1), 6

Heyman J. The stone skeleton. Int J Solids Struct. 1966;2:249–52

2.3.    Methods
*       Methodology limitations is needed.
*       I would recommend presenting the methodology through a flowchart.
*       Future research and recommendation section are missing.
*       Please improve all figures, in terms of text sizes, resolution, present in a more professional way.
2.4.    Results
*       Please present some results through graphs.

2.5.    Discussion
In general, this section is well written. Nevertheless, the discussion should provide a summary of the main finding(s) of the manuscript in the context of the broader scientific literature, as well as addressing any limitations of the study or results that conflict with other published work.

Please put some quantitative findings in the conclusion section.

2.6.    References
*       Please check the references in the text and the list; some of them are not according to the journal style.

Author Response

Dear reviewer,

Please find attached the response to your comments.

Sincerely

Reviewer 3 Report

Dear editors,

The paper from Ribeiro et al. deals with the terracotta marks found in Braga (Portugal). The paper objective is “to correlate, even statically, metrological data, typologies, and technologies with archaeological and historical evidence.” The material is abundant and surely complex, the effort to systematize it is clear. However, the aims proposed are only partially reached. The paper is indeed the first typological classification on the marks, but all the other aspects are missing from the paper.  Statistical treatment of data goes beyond presenting a frequency of each mark and could for example tell us if some signs are correlated or not to specific terracotta types, or contexts or chronologies. This last element is almost bypassed by authors who reach conclusions of the diversity of marks not considering different phases of production as a factor. The authors might find examples of statistical treatments of these data in studies of decorative motives in ceramics. In conclusion, the paper fails to tell the reader basic but important information about the terracotta production in Braga. I suggest to authors rephrase the paper by clearly stating that is a first typological classification, to focus on aspects like chronology and context distribution and leave all the other aspects for further work.  As the rewriting is extensive, I suggest the rejection of the paper in its actual state and its resubmission. I wish the authors good luck with their work.

The Introduction and paragraphs 2 and 3 need to be cut, synthesized and merged: much information is repeated making the reading difficult.

Sample paragraph: infos are sequenced clumsily. It´s hard to get the most fundamental infos that are: number of samples, contexts and chronology.

Figures 1-2 are not cited in the text and needs to be removed.

The text is readable but the language needs extensive review, as the word choice is often incorrect. The wrong words are often just highlighted.

More detailed comments on the text

Author Response

(The authors gave the same response as above.)

Reviewer 4 Report

This interesting works deals with a kind of material that has been largely unexplored in academic works: ATC architectural terra-cotta (tiles, bricks and pipes…). In this work, authors have analyzed 1216 marks, from 41 archaeological sites, 19 suggested an organized and dynamic production, and an open-market, supported by numerous 20 officinae, of different sizes, some of them located near the housing area, and the presence of a large 21 number of workers, including women and children. Therefore, present study is a deep work on the ATC materials diachronically from 1st century BC to the 5th century AD and from 41 sites. It is well written and data supported. It deserves publication after some improvements.

Figure 1. mark or explain in the figure caption which are the Roman low-imperial walls and the medieval walls. It is not clear what are the squares depicted in the plan (insulae?) and why are depicted over the city walls. Is the darker wall the medieval one?

Line 46: 1th is 1st instead

Better use superindex notation for ordinal numbers

There are several paragraphs with just one sentence. Please, merge these paragraphs with the one before or one after following the same idea for better readability. For instance, lines 63-71.

Better change word “oven” for “kiln” (line 169).

Line 198 and following: Section 5. Methods. Please, explain and discuss about the methodology proposed by Goulpeau and Le Ny and followed in this work. Otherwise, reader cannot fully understand the methodology adopted by authors in this comprehensive study. How does this methodology correlate with the protocol proposed by ATC Network?

It could be interesting showing the ratio of appearance of each mark in view of each ATC typology. In table 2 authors collect and show the different ATC typologies and the number of marks, calculating the percentage of abundance for each marked typology vs the total of the statistical sample. Besides, in Table 3, authors show the frequency of the different marks. However, to get a more comprehensive study, it is suggested crossing both types of data (marks and ATC typologies). Doing so maybe some trend can be identified in the use of marks for specific ATC material, although the sampling might be biased in the past (as noted by authors regarding Carvalheiras domus).

Author Response

(The authors gave the same response as above.)

Round 2

Reviewer 2 Report

The present state of the article now is acceptable for publishing in Heritage.

Reviewer 3 Report

Dear authors,

the text has been extensively edited, it reads well, and the goals of the papers are accomplished. Still, language needs further improvements as a few words are used in a wrong way, i.e. ceramic consolidation phase instead of ceramic drying phase, an issue which needs to be addressed. I believe that the chronology is a key factor which needs to be addressed in order to give some thoughtful conclusions of the topic; however, now it is clear that the paper is an initial study on the material. With the correction suggested in the pdf attached, the paper can be published in the journal.

Best wishes,
